# Is the Addition of Chemotherapy to Adjuvant Radiation in Merkel Cell Cancer Beneficial? Real-World Data with Long-Term Follow-Up

**DOI:** 10.3390/cancers17060945

**Published:** 2025-03-11

**Authors:** Walid Shalata, Hanna T. Frumin Edri, Ina Sarel, Anna Ievko, Sofiia Turaieva, Tanzilya Tairov, Ilia Berezhnov, Shlomit Fenig, Eyal Fenig, Tomer Ziv-Baran, Alexander Yakobson, Ronen Brenner

**Affiliations:** 1The Legacy Heritage Cancer Center, Dr. Larry Norton Institute, Soroka Medical Center, Beer Sheva 84105, Israel; walid_sh@clalit.org.il (W.S.); alexy@clalit.org.il (A.Y.); 2Faculty of Health Sciences, BenGurion University of the Negev, Beer Sheva 84105, Israel; 3Edith Wolfson Medical Center, Oncology Institute, Holon 5822012, Israel; hanafr@wmc.gov.il (H.T.F.E.); annai@wmc.gov.il (A.I.); sofiiat@wmc.gov.il (S.T.); tanzilyat@wmc.gov.il (T.T.); ilyaberezhnov4@gmail.com (I.B.); 4Independent Researcher, Ramat Gan 5241611, Israel; ina@nurexone.com; 5Institute of Oncology, Kaplan Medical Center, Faculty of Medicine, Hebrew University, Jerusalem 9190500, Israel; shlomitfen@clalit.org.il; 6Institute of Oncology, Davidoff Center, Rabin Medical Center, Beilinson Hospital, Petah Tikva 4941492, Israel; efenig@clalit.org.il; 7School of Public Health, Sackler Faculty of Medicine, Tel-Aviv University, Tel Aviv 6997801, Israel; zivtome@tauex.ac.il

**Keywords:** Merkel cell carcinoma, adjuvant therapy, radiation therapy, chemotherapy, survival analysis, long-term outcomes

## Abstract

Merkel cell carcinoma (MCC) is a rare and aggressive malignancy of the skin whose primary treatment modalities for non-metastatic disease are surgery and radiation. In cases of resectable MCC, the addition of concurrent chemotherapy to adjuvant radiation therapy is still debatable. According to this real-world study, adding chemotherapy to adjuvant radiation does not significantly improve overall or disease-free survival in these patients. However, even though their disease was significantly more advanced, the chemoradiation-treated group had similar survival outcomes. This could imply that some high-risk subgroups may benefit from this more aggressive strategy.

## 1. Introduction

Merkel cell carcinoma (MCC) is a rare and aggressive neuroendocrine skin cancer with increasing incidence over the past two decades [1,2,3,4]. Despite its rarity, with approximately 1600 cases per year in the United States, the health impact of MCC has grown significantly, with its reported incidence tripling between 1986 and 2001 [5,6,7,8]. This increase may be partially attributed to improved diagnostic techniques, particularly the introduction of cytokeratin-20 staining, but also reflects the rising prevalence of known risk factors, including T-cell immune suppression, advanced age, and extensive UV exposure in fair-skinned individuals [9,10,11,12]. The optimal management of MCC, particularly regarding adjuvant therapy, remains controversial [9]. While surgery is the primary treatment for localized disease, the high risk of recurrence has led to the routine consideration of adjuvant therapies [13,14,15,16]. Multiple studies have demonstrated improved outcomes with adjuvant radiation therapy (RT) in early-stage disease [17,18,19,20,21]. A landmark National Cancer Database (NCDB) analysis of 6908 patients showed that adjuvant RT was associated with improved overall survival in stage I-II MCC (hazard ratio [HR] = 0.71, 95% CI = 0.64–0.80, *p* < 0.001 for stage I; HR = 0.77, 95% CI = 0.66–0.89, *p* < 0.001 for stage II) [14]. However, the role of adding chemotherapy to adjuvant radiation remains unclear [22]. While chemotherapy can enhance cell death caused by radiation treatment by acting as a radiosensitizing agent, previous studies examining the benefit of adjuvant chemotherapy have yielded conflicting results [18,23]. In a relatively small study of 37 patients, the results supported the use of combined treatment with chemotherapy followed by radiation therapy for patients with advanced locoregional Merkel cell carcinoma [24]. A prospective phase II trial by Poulsen et al. comparing synchronous carboplatin/etoposide with radiation versus historical controls treated with radiation alone found no significant survival benefit from the addition of chemotherapy in high-risk stage I and II disease [25]. An extensive NCDB analysis of 4815 patients showed that while postoperative chemoradiation improved overall survival compared to radiotherapy alone, this benefit was primarily limited to specific high-risk subgroups, such as patients with positive margins or tumors ≥ 3 cm and male sex [16].

The question of whether chemotherapy adds value to adjuvant radiation therapy in resectable MCC remains particularly relevant given the potential toxicities and immune suppressive effects of cytotoxic chemotherapy in a disease where immune function appears to play a crucial role in outcomes [26,27,28]. We conducted a multi-center analysis with 20 years of follow-up to address this clinical question. We examined the outcomes of 105 patients treated with surgery followed by either radiation alone or chemoradiation. Our study represents one of the most extended follow-up periods reported in the literature for MCC treatment outcomes.

## 2. Materials and Methods

### 2.1. Study Design and Participants

This retrospective multicenter cohort study analyzed 105 patients diagnosed with local or locally advanced MCC who underwent surgical excision followed by adjuvant therapy at three major medical centers in Israel between September 1985 and February 2024. The study was approved by the Institutional Review Boards of all participating centers: Rabin Medical Center, Edith Wolfson Medical Center, and Soroka Medical Center.

### 2.2. Data Collection

Patients were identified through a systematic review of pathology and oncology department records using standardized diagnostic codes. Data were extracted from electronic medical records using a standardized collection form by trained research staff. The variables collected included patient demographics (age at disease onset and gender), tumor characteristics (location), and treatment details (surgery, chemotherapy and radiation protocols, and amount of radiation received). The two treatment groups (chemoradiation vs. radiation only) were treated during the same time periods. Radiotherapy was performed according to similar protocols and standards of care.

Clinical staging was determined according to the primary tumor (T), the presence of positive lymph nodes (N), and the presence of distant metastases (M) at the time of first diagnosis (TNM) according to the classification described in the American Joint Committee on Cancer (AJCC) staging system [29]. Patients with insufficient records or no follow-up data were excluded from the analysis.

The clinical outcomes were overall survival (OS) and disease-free survival (DFS). OS was calculated as the duration from the diagnosis of MCC to any cause of death or the last follow-up. DFS was calculated from MCC diagnosis until recurrence of the disease or death or last follow-up.

Death records were obtained from the Israel Ministry of Interior. The follow-up was censored on 1 February 2024.

### 2.3. Statistical Analysis

Categorical variables were summarized as frequencies and percentages. The distribution of continuous variables was evaluated using histograms and Q-Q plots. Continuous variables were reported as the median and interquartile range (IQR). An independent samples *t*-test and a Mann–Whitney test were used to compare continuous variables between treatment groups, while categorical variables were compared using the Chi-square test or Fisher’s exact test. The reverse censoring method was used to evaluate the median length of follow-up. Survival analysis was conducted using Kaplan–Meier curves to describe mortality and disease-free survival in each group, with log-rank tests used for between-group comparisons. Multivariable Cox regression was applied to study the association between chemotherapy use and patient survival while controlling for potential confounders. Variables that were significantly different between groups were considered potential confounders. All statistical tests were two-sided, with *p* < 0.05 considered statistically significant. The analyses were performed using SPSS software (IBM SPSS version 28, Armonk, NY, USA, 2021).

## 3. Results

### 3.1. Study Population

A total of 105 patients with resectable Merkel Cell Carcinoma received adjuvant therapy following surgery, with 52 patients receiving combined chemoradiation and 53 receiving radiation alone. The study included 833 total person-years of observation (median length of follow-up of 12 years). The demographic and clinical characteristics differed between the treatment groups (Table 1). Patients in the chemoradiation group were significantly younger (median age 65.9 [IQR 59–73.0] vs. 77.3 [IQR 61.4–83.3] years, *p* = 0.002) and received higher radiation doses (median 50 Gy [IQR 45–50] vs. 45 Gy [IQR 38–50], *p* = 0.002). While male predominance was observed in both groups (69.2% vs. 56.6%), this difference was not statistically significant (*p* = 0.181).

### 3.2. Sample Size

The sample size was calculated using a significance level of 5% and power of 80%, assuming the same sample size in both groups. We assumed that the average probability of survival to the end of follow-up (20 years) was 30%. Using a hazard ratio (HR) of 0.5, 102 patients were needed, i.e., 51 in each group.

### 3.3. Disease and Treatment Characteristics

The primary tumor location varied between groups, with the head and neck being the most commonly identified primary site in both groups (25% vs. 28.3%, *p* = 0.702). Notably, non-metastatic unknown primary tumors were significantly more frequent in the chemoradiation group (34.6% vs. 1.9%, *p* < 0.001).

The disease stage characteristics showed essential differences between groups. The chemoradiation group presented with a more advanced T stage (*p* = 0.035), with 60.6% having T2-T4 disease compared to 40.4% in the radiation-alone group. Nodal involvement was also significantly more common in the chemoradiation group. Consequently, TNM staging showed significant differences (*p* < 0.001), with 88% of chemoradiation patients presenting with stage III disease compared to 28.3% in the radiation-alone group (Table 1).

The median radiation dose was lower in the radiation-only group compared to the chemoradiation group (45 (38–50) GY vs. 50 (45–50) GY, respectively).

All chemotherapy patients included in this study received platinum and etoposide.

### 3.4. Survival Outcomes

During the follow-up period, 22 patients died in the chemoradiation group and 28 in the radiation-only group. The 20-year survival rate was 53.4% in the chemoradiation group compared to 30.7% in the radiation-only group (*p* = 0.324) (Figure 1). Median survival in the chemoradiation group was not reached during the follow-up period; in the radiation group, it was 8.8 years. Since 20-year survival is a long time period, especially for the age group of our patients, we also calculated the 5-year OS rate. The 5-year OS rates were quite similar between the groups: 64.9% in the chemoradiation group and 64.5% in the radiation group (*p* = 0.89). In a multivariate logistic regression analysis including age at diagnosis, TNM stage, and disease location, chemoradiotherapy was not associated with a significant difference in mortality for either 5-years ((HR = 1.547, 95% CI 0.656–3.65, *p* = 0.319) or 20-year OS (HR = 1.358, 95% CI 0.614–3.003, *p* = 0.450) (Table 2). Unlike age at diagnosis, stage and tumor location did not influence mortality in the multivariate analysis (Table 2).

### 3.5. Disease-Free Survival

During follow-up, 56 patients out of the 105 experienced disease recurrence or death, 26 in the chemoradiation group and 30 in the radiation-only group. The 20-year disease-free survival rate was not significantly different between groups: 47% in the chemoradiation group versus 29.3% in the radiation-only group (*p* = 0.495). The five-year disease-free survival rates were also comparable, 57.4% in the chemoradiation group and 58.7% in the radiation group (*p* = 0.80). Similarly to overall survival, the multivariate analysis at 5 years of DFS, including age at diagnosis, TNM stage, and disease location, indicated no significant difference in DFS between treatment groups (HR = 1.324, 95% CI 0.592–2.960, *p* = 0.494) (Table 3). The 5- and 20-year DFS Kaplan–Meier (KM) curves are presented in Appendix A.

### 3.6. Survival Outcomes in MCC Patients Excluding Unknown Primary

Merkel cell carcinoma (MCC) of unknown primary origin is known to have a better prognosis compared to MCC with a known primary site. Since many patients in the chemoRT group had stage III disease of unknown primary, this may have biased the survival results in favor of chemoRT. To address this, we conducted a survival analysis excluding these patients. The survival rates remained not statistically different between the two groups: the 20-year overall survival rates were 51.6% for chemoradiation versus 31.2% for radiation alone (*p* = 0.58). The survival rates after only 5 years of follow-up were also not different: 60.8% in the chemoradiation group and 65.8% in the radiation group (*p* = 0.54) (Table 4).

Likewise, the twenty-year disease-free survival (DFS) rates were not significantly different between chemoradiation and radiation: the 20-year DFS rates were 46.7% in the chemoradiation group vs. 29.9%, respectively, *p* = 0.62. The five-year DFS rates were 55.3% in the chemoradiation group and 59.9% in the radiation group (*p* = 0.64) (Table 4). The 5- and 20-year DFS Kaplan–Meier (KM) curves are presented in Appendix A.

## 4. Discussion

This multi-center study, with a long reported follow-up period in MCC treatment (833 person-years, median 12 years), provides important insights into the role of adjuvant chemoradiation versus radiation alone in resectable MCC.

Firstly, in our cohort, the addition of chemotherapy to adjuvant radiation did not significantly improve overall survival or disease-free survival. Even after adjusting for significant predictive variables, including age, disease stage, and tumor site, the survival rates in the two groups remained comparable. However, several key observations emerge from our analysis. First, this is a real-world retrospective study, and the adjuvant treatments were given at the physician’s discretion. We assume this was the probable clinical rationale for the significantly different number of patients with stage III in the chemoradiation group (88% vs. 28.3%), i.e., a possible preference for three-modality treatment in this high-risk population. Nevertheless, despite this fundamental adverse prognostic factor, their survival outcomes were comparable to those of the radiation-only group (53.4% vs. 30.7% in the chemoradiation and radiation-only groups, respectively (*p* = 0.324)).

Chemoradiation was probably preferred for younger patients due to its higher toxicity, as expressed by the significant difference in median age (median age 65.9 vs. 77.3 years in chemoradiation vs. radiation, respectively). According to the same reasoning, patients in the chemoradiation group received higher radiation doses (median 50 Gy vs. 45 Gy). The higher radiation doses in patients receiving chemoradiation may have influenced the outcome. Nevertheless, although significant, the actual difference between median radiation doses was relatively small. Thus, this difference was likely less influential on the overall outcomes compared to the effects of chemotherapy.

It may have been anticipated that these treatment factors would favor better outcomes in this group. To overcome the differences in the essential characteristics of the patients, we performed a multivariate analysis including chemotherapy treatment and the staging, age, and location of the primary tumor. Nevertheless, survival was still not statistically different between the groups.

The fact that patients with significantly more advanced diseases achieved similar survival outcomes raises the possibility that chemotherapy might provide benefits in high-risk patients, probably neutralizing their worse prognosis.

Our results align with and challenge previous studies examining the value of adjuvant chemotherapy in MCC. The prospective trial by Poulsen et al. [25] comparing synchronous carboplatin/etoposide with radiation versus historical controls found no survival benefit from chemotherapy addition in high-risk stage I and II disease. Similarly, the extensive NCDB analysis by Bhatia et al. [14] of 6908 cases showed no survival benefit of adjuvant chemotherapy in stage III disease (HR = 0.97, 95% CI = 0.85–1.12, *p* = 0.71).

However, an NCDB analysis of 4815 patients showed that postoperative chemoradiation may improve overall survival compared to radiotherapy alone in males and in specific high-risk subgroups, such as patients with positive margins or tumors ≥ 3 cm [16].

Thus, our observation of similar survival outcomes despite significantly more advanced disease in the chemoradiation group may also suggest a potential role for chemotherapy in selected high-risk patients [15,24].

Our study has several strengths, including its long follow-up period, multi-center design, and detailed patient characterization. The median follow-up of 12 years allows for a meaningful assessment of long-term outcomes, as late recurrences can occur in MCC [30]. However, we acknowledge several limitations. The retrospective nature of the study limits the possibility of collecting additional data. In this observational research, the treatment groups have significantly different baseline characteristics. While we attempted to control for these differences through multivariate analysis, unmeasured confounders may exist. An additional limitation is the lack of data on specific mortality, comorbidities, and serious side effects.

Our study spans from 1985 to 2023, a period during which MCC metastatic management has dramatically changed. Chemotherapy use in metastatic patients has declined significantly in favor of immunotherapy in recent years [31].

Immunotherapy was approved for MCC in Israel from 2018, and only for metastatic MCC. Since the patient population in this study had local/locally advanced MCC, the only patients who could have benefited from immunotherapy were those who had disease recurrence after 2018. In this period, there were three recurrences in the chemoradiation group and four in the radiation-only group. We do not have information on whether their disease recurrence was metastatic. Nevertheless, the small number of patients and the same distribution between the groups suggest that immunotherapy could not have influenced the outcome of this study.

These findings have important clinical implications and suggest that certain high-risk subgroups might benefit from this more aggressive approach. This is particularly relevant given the emergence of immunotherapy as a promising treatment option in MCC, with recent studies showing encouraging results with checkpoint inhibitors; nevertheless, not all patients are eligible for or responsive to immunotherapy [27,28].

In metastatic lung cancer, particularly in metastatic small-cell lung carcinoma (SCLC), an aggressive, high-grade neuroendocrine carcinoma resembling MCC, immunotherapy has transformed the treatment paradigm. Moreover, it was found that the addition of immunotherapy to chemotherapy significantly improved the outcome [32]. Since MCC is also an aggressive disease with high mortality and, unlike melanoma, is responsive to chemotherapy, it may be logical to assume that combining chemotherapy with immunotherapy in MCC may improve survival in metastatic disease. It is possible that in the future, chemotherapy’s combined radiation and immunotherapy will also be beneficial in an adjuvant setting. Thus, future prospective studies should focus on identifying subgroups that might benefit from combined intensified therapy in different stages in this rare and very aggressive disease.

## 5. Conclusions

While our multivariate analysis showed no overall survival benefit from adding chemotherapy to adjuvant radiation, the similar survival outcomes, despite significantly more advanced disease in the chemoradiation group, suggest a potential benefit in high-risk patients. These findings warrant further investigation through prospective studies with larger, stage-matched cohorts to definitively establish the role of adjuvant chemotherapy in high-risk resectable MCC. The combination of chemotherapy and immunotherapy may have a synergistic effect on survival, similarly to the case of SCLC, also a high-grade neuroendocrine carcinoma. This hypothesis should be tested in future prospective studies in appropriately characterized patients with metastatic disease and possibly also with the combination of chemoradiation and immunotherapy in neoadjuvant and adjuvant settings.

## Figures and Tables

**Figure 1 cancers-17-00945-f001:**
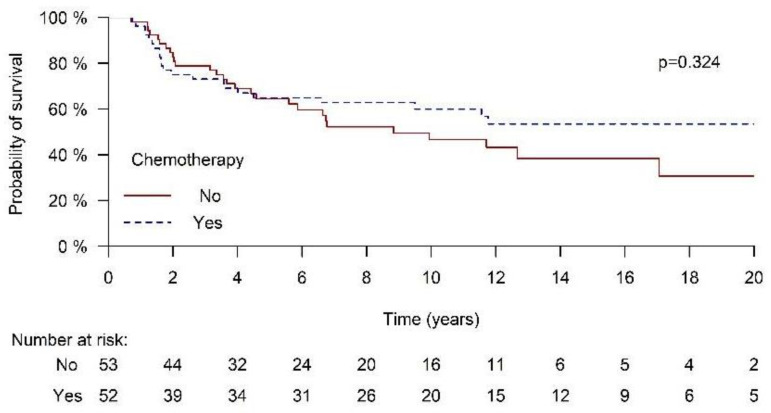
Overall survival in radiation- and chemoradiation-treated MCC patients.

**Table 1 cancers-17-00945-t001:** Patient and disease characteristics by treatment group.

Characteristic	Chemoradiation (*n* = 52)	Radiation Only (*n* = 53)	*p* Value
Age, median (IQR [years])	65.9 (59–73)	77.3(61.4–83.3)	0.002
Male (%)	36 (69.2)	30 (56.6)	0.181
Radiation (GY) median (IQR) [GY]	50 (45–50)	45 (38–50)	0.002
Tumor site			
Head and Neck	13 (25)	15 (28.3)	0.7
Torso	5 (9.6)	7 (13.2)	
Upper limbs	3 (5.8)	14 (26.4)	
Lower limbs	6 (11.5)	12 (22.6)	
Buttocks	7 (13.5)	14 (7.5)	
Unknown primary (lower body)	15 (28.8)	1 (1.9)	
Unknown primary (upper body)	3 (5.8)	0	
T staging			0.035
T1	13 (39.4)	31 (59.6)	
T2	15 (45.5)	18 (34.6)	
T3	4 (12.1)	3 (5.8)	
T4	1 (3)	0 (0)	
N staging			<0.001
N0	6 (11.5)	37 (71.2)	
N1	43 (82.7)	14 (26.9)	
N2	3 (5.8)	1 (1.9)	
TNM Stage			<0.001
1	3 (5.8)	24 (45.3)	
2a	2 (3.8)	14 (26.4)	
2b	1 (1.9)	0	
3a	16 (30.8)	11 (20.8)	
3b	30 (57.2)	4 (7.5)	

**Table 2 cancers-17-00945-t002:** Multivariate analyses of 5- and 20-year overall survival.

	OS-5 Years			OS-20 Years		
Variable	HR	95% CI	*p* Value	HR	95% CI	*p* Value
Chemotherapy	1.547	0.656–3.65	0.319	1.358	0.614–3.003	0.450
Age at diagnosis	1.064	1.029–1.100	<0.001	1.069	1.039–1.101	<0.001
TNM stage			0.696			0.722
I	Ref *			Ref *		
II	1.114	0.385–3.229		0.696	0.283–1.713	
III	1.492	0.570–3.904		0.827	0.373–1.833	
Location			0.113			0.528
Head and neck	Ref *			Ref *		
Rest of body	0.452	0.213–0.960		0.688	0.360–1.315	
Unknown primary	0.517	0.173–1.546		0.781		

*—Reference group.

**Table 3 cancers-17-00945-t003:** Multivariate analyses of 5-year disease-free survival.

	DFS-5 Years		
Variable	HR	95% CI	*p* Value
Chemotherapy	1.324	0.592–2.960	0.494
Age at diagnosis	1.048	1.018–1.078	<0.001
TNM stage			0.819
I	Ref *		
II	1.229	0.459–3.293	
III	1.336	0.539–3.311	
Location			0.17
Head and neck	Ref *		
Rest of body	0.517	0.260–1.217	
Unknown primary	0.685	0.263–1.784	

*—Reference group.

**Table 4 cancers-17-00945-t004:** Survival outcomes in MCC patients excluding unknown primary.

	Chemoradiation (*n* = 34)	Radiation (*n* = 34)	*p* Value
5-Year OS (%)	60.8	65.8	0.54
20-Year OS (%)	51.6	31.2	0.58
5-Year DFS (%)	55.3	59.9	0.64
20-Year DFS (%)	46.7	29.9	0.62

## Data Availability

The data presented in this study are available on request from the corresponding authors.

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
