# Peer review of "Is the Addition of Chemotherapy to Adjuvant Radiation in Merkel Cell Cancer Beneficial? Real-World Data with Long-Term Follow-Up"

_cancers, 2025, doi:10.3390/cancers17060945_

Round 1
Reviewer 1 Report
Comments and Suggestions for Authors
Summary
This study evaluates the role of adding chemotherapy to adjuvant radiation therapy (chemoRT) in resectable MCC through a retrospective multicenter cohort analysis. The authors conclude that while the survival benefit of chemoRT over RT alone was not statistically significant, the comparable survival despite a more advanced disease stage in the chemoRT group suggests a potential benefit for high-risk patients.
A major strength of this study is its exceptionally long follow-up period, spanning multiple decades (median 12 years, maximum 20 years). This allows for an assessment of long-term survival trends. However, given the evolving landscape of MCC treatment over this period, including the introduction of immunotherapy, there are critical confounders that may impact the interpretation of results.
Major Comments
- Potential Temporal Confounders
- The study spans from 1985 to 2023, a period during which MCC management has dramatically changed. Most notably, chemotherapy use has declined significantly in favor of immunotherapy in recent years. At the same time, advances in surgical techniques, radiation therapy, and overall patient management have likely improved survival rates. It is important to assess whether the two treatment groups (chemoRT vs. RT monotherapy) were treated during different time periods, as this could confound the results. The authors should provide a breakdown of treatment year distribution and consider adjusting for this factor in their analysis.
- Favorable Prognosis of MCC of Unknown Primary as a Confounder
- MCC of unknown primary is well recognized to have a better prognosis than MCC with a known primary. Given that a large proportion of patients in the chemoRT group had stage III disease of unknown primary, this may have skewed survival in favor of chemoRT. The authors should either adjust for this factor in their analysis or discuss its potential impact in the limitations section.
- Delayed Divergence in Kaplan-Meier Curves
- The Kaplan-Meier survival curves for OS show a divergence between chemoRT and RT groups only after six years. If chemotherapy were improving MCC-specific outcomes, one would expect to see a separation earlier in the course of follow-up due to reduced recurrence rates. This delayed divergence suggests that other factors, such as baseline patient health or long-term treatment effects, may be influencing survival. The authors should explore this in more detail and consider analyzing competing risks (e.g., deaths due to non-MCC causes). Simply, chemoRT patients may have been healthier in this graph.
- Chemotherapy-Induced Response vs. Significant Toxicity
- While chemotherapy may enhance tumor response or act as a radiosensitizer, it is also associated with significant toxicity, particularly in elderly patients who represent a large proportion of the MCC population. The authors may consider including a detailed breakdown of serious side effects in both cohorts.
- Identification of High-Risk Subsets for ChemoRT
- The authors suggest that chemoRT might still be beneficial for certain high-risk patients. If this is the case, it is importnt to specify the exact risk factors that should guide the use of chemoRT. For example, did patients with large primary tumors, extracapsular nodal extension, or positive margins derive greater benefit? Clearly defining these characteristics based on the data will provide actionable clinical insights and improve the study’s impact.
- Integration of Chemotherapy with Immunotherapy
- Immunotherapy is now the standard first-line systemic therapy for MCC, drastically changing the treatment paradigm. The authors should discuss how their findings should be interpreted in this new era. For instance, should chemotherapy still be considered in combination with immunotherapy for high-risk patients, or is its role now obsolete?
- Relevance of Chemotherapy in the Immunotherapy Era: While chemotherapy, particularly in the adjuvant setting, is becoming less relevant in MCC due to the growing use of immunotherapy, the data presented in this study may still provide valuable insights for the clinical and research community. However, the discussion should more prominently emphasize the ongoing investigations into adjuvant immunotherapy. The authors may address how their findings should be interpreted in the context of immunotherapy trials and whether chemotherapy has any residual role in specific high-risk subsets.
- Significant Age Disparity Between Groups and Its Impact on OS: The chemoRT group was approximately 12 years younger than the RT-only group. Given that younger age is a strong independent prognostic factor, this imbalance could have significantly influenced the observed survival trends. Disease-specific survival may provide a more precise understanding of the impact of chemoRT on MCC outcomes. If possible, the authors should include DSS data, with K-M curve, as it would help distinguish deaths due to MCC from other causes, particularly given the older age of the RT-only group.
Minor Comments
- Interpretation of Multivariable Analysis in Table 2
- The multivariable analysis in Table 2 suggests that stage did not significantly impact OS (HR=0.7, p=0.43). This finding is unexpected, as prior studies have consistently shown that advanced stage is a major prognostic factor in MCC. If my understanding is correct, this discrepancy raises concerns about the validity of the model and should be discussed.
- Details on Chemotherapy Regimens and ChemoRT Schedule
- The manuscript does not specify which chemotherapy regimens were used or the timing of chemotherapy administration relative to radiation. Were chemotherapy and radiation given concurrently, sequentially, or variably across patients?
Author Response
Ronen Brenner, MD
Director of the Oncology Institute
Wolfson Medical Center
62 Halohamim Street, Holon 5822012, Israel
ronenbr@wmc.gov.il
(972)-54-3456721
28 February, 2025
Editor-in-Chief
Cancers
cancers@mdpi.com
Dear Editor and Reviewers,
Thank you for thoroughly reviewing our manuscript “Is the addition of chemotherapy to adjuvant radiation beneficial in Merkel cell cancer? A real-world data analysis with long-term follow-up”. We greatly appreciate your dedicated time and effort in providing comprehensive feedback. Your constructive comments and suggestions have greatly enhanced the content of our manuscript. We have carefully addressed all the points raised and revised our manuscript accordingly. Below, we provide a point-by-point response to each comment, with detailed explanations of the changes made. The modifications are also highlighted in the revised manuscript.
Response to Reviewer #1:
- Potential Temporal Confounders
The study spans from 1985 to 2023, a period during which MCC management has dramatically changed. Most notably, chemotherapy use has declined significantly in favor of immunotherapy in recent years. At the same time, advances in surgical techniques, radiation therapy, and overall patient management have likely improved survival rates. It is important to assess whether the two treatment groups (chemoRT vs. RT monotherapy) were treated during different time periods, as this could confound the results. The authors should provide a breakdown of treatment year distribution and consider adjusting for this factor in their analysis.
Response:
We would like to thank the reviewer for this valuable comment, which was also addressed in the revised manuscript (lines 289-296). We agree with the reviewer that immunotherapy was the most important transformation in MCC’s management during the study span.
Immunotherapy was introduced for MCC in the Israeli Health Benefits Package in 2018 and only for metastatic MCC. Since the patient population in this study had local MCC, the only patients who could have benefited from immunotherapy were those who had disease recurrence after 2018. There were seven such patients with recurrent disease during 2018-2023, with a similar distribution between the groups, three from the chemoradiation group and four from the radiation-only group. We do not have information on whether their disease recurrence was metastatic. Nevertheless, the small number of patients and the same distribution between the groups suggest that immunotherapy could not have influenced the outcome of this study.
The two groups, chemoradiation and radiation only, received treatment evenly during the same study years. Chemotherapy and radiation were administered during the study using the same standard of care and regimen. The drugs used for all chemotherapy patients included in the study were platinum and etoposide.
- Favorable Prognosis of MCC of Unknown Primary as a Confounder
MCC of unknown primary is well recognized to have a better prognosis than MCC with a known primary. Given that a large proportion of patients in the chemoRT group had stage III disease of unknown primary, this may have skewed survival in favor of chemoRT. The authors should either adjust for this factor in their analysis or discuss its potential impact in the limitations section.
Response:
We agree with the reviewer and thank him for this good comment. We have followed the reviewer’s suggestion and made an additional analysis without unknown primary patients and included it in the revised manuscript (lines 211-228). The survival rates remained not statistically different between the two groups: The 20-year overall survival rates were 51.6% for chemoradiation versus 31.2% for radiation alone (p=0.58) (see figure 1 below). The survival rates after only 5 years of follow-up were also not different: 60.8% in the chemoradiation group and 65.8% in the radiation group (p=0.54) (figure 1 and supplement 1).
Figure 1: 5 and 20-year Overall Survival in Radiation and Chemoradiation Treated MCC Patients excluding unknown primary
Likewise, twenty-year disease-free survival (DFS) rates were not significantly different between chemoradiation and radiation: 20-year DFS rates were 46.7% in the chemoradiation group vs. 29.9%, respectively, p=0.62 (figure 2). Five-year DFS rates were 55.3% in the chemoradiation group and 59.9% in the radiation group (p=0.64) (figure 2 and in supplement 1).
Figure 2: 5 and 20-year Disease-free Survival in Radiation and Chemoradiation Treated MCC Patients excluding unknown primary
- Delayed Divergence in Kaplan-Meier Curves
The Kaplan-Meier survival curves for OS show a divergence between chemoRT and RT groups only after six years. If chemotherapy were improving MCC-specific outcomes, one would expect to see a separation earlier in the course of follow-up due to reduced recurrence rates. This delayed divergence suggests that other factors, such as baseline patient health or long-term treatment effects, may be influencing survival. The authors should explore this in more detail and consider analyzing competing risks (e.g., deaths due to non-MCC causes). Simply, chemoRT patients may have been healthier in this graph.
Response:
We agree with the reviewer that chemoRT patients may have been healthier (in respect with comorbidities) but they also had more advanced disease (12.1% T3 vs. 5.8% T3 in the RT group). There was insufficient data to analyze the subgroups; however, it is established that the primary factor influencing patients' survival is the severity of the disease. Although the graphs showed separation at later stages, this difference was not statistically significant. We did not assert that patients receiving chemoradiotherapy (chemoRT) had better survival rates; we reported similar survival rates despite their more advanced disease. Therefore, we concluded that chemotherapy was possibly beneficial for these patients.
- Chemotherapy-Induced Response vs. Significant Toxicity
While chemotherapy may enhance tumor response or act as a radiosensitizer, it is also associated with significant toxicity, particularly in elderly patients who represent a large proportion of the MCC population. The authors may consider including a detailed breakdown of serious side effects in both cohorts.
Response:
We thank the reviewer for this comment. Unfortunately, data on serious side effects was not available.
- Identification of High-Risk Subsets for ChemoRT
The authors suggest that chemoRT might still be beneficial for certain high-risk patients. If this is the case, it is importnt to specify the exact risk factors that should guide the use of chemoRT. For example, did patients with large primary tumors, extracapsular nodal extension, or positive margins derive greater benefit? Clearly defining these characteristics based on the data will provide actionable clinical insights and improve the study’s impact.
Response:
We sincerely thank the reviewer for their insightful comments. Based on our findings, we concluded that chemotherapy might be beneficial for patients with more advanced disease, particularly for younger individuals who may better tolerate the potential toxic effects of chemotherapy. However, due to the retrospective nature of our study and limitations in the resolution of our patients' data, we were unable to establish a precise risk stratification index. This may be addressed in future prospective studies.
- Integration of Chemotherapy with Immunotherapy
Immunotherapy is now the standard first-line systemic therapy for MCC, drastically changing the treatment paradigm. The authors should discuss how their findings should be interpreted in this new era. For instance, should chemotherapy still be considered in combination with immunotherapy for high-risk patients, or is its role now obsolete?
Response:
Thank you for this comment that enhances our manuscript’s content and was added to discussion in lines 302-310 and 319-324. In metastatic lung cancer and particularly in metastatic small cell lung carcinoma (SCLC), another aggressive, high-grade neuroendocrine carcinoma resembling MCC, immunotherapy also transformed the treatment paradigm. Moreover, it was found that combining chemotherapy with immunotherapy significantly improved the outcome. Since MCC is also an aggressive disease with high mortality and unlike melanoma, responsive to chemotherapy, it may be logical to assume that adding chemotherapy to immunotherapy may improve survival in metastatic disease. It is possible that in the future, chemotherapy’s addition to immunotherapy will be beneficial also in the adjuvant setting.
- Relevance of Chemotherapy in the Immunotherapy Era: While chemotherapy, particularly in the adjuvant setting, is becoming less relevant in MCC due to the growing use of immunotherapy, the data presented in this study may still provide valuable insights for the clinical and research community. However, the discussion should more prominently emphasize the ongoing investigations into adjuvant immunotherapy. The authors may address how their findings should be interpreted in the context of immunotherapy trials and whether chemotherapy has any residual role in specific high-risk subsets.
Response:
We completely agree with the reviewer. Immunotherapy, which is still primarily used in MCC metastatic disease, will be used much more in the future as neoadjuvant and probably adjuvant treatment. Since similarly to SCLC, MCC is an aggressive high-grade neuroendocrine tumor, chemotherapy’s role in MCC is not obsolete; on the contrary, it may be used in addition to immunotherapy to improve survival. Furthermore, in patients with contraindications for immunotherapy and patients with local recurrent disease, the use of chemotherapy concurrently with radiation can still be an option.
- Significant Age Disparity Between Groups and Its Impact on OS: The chemoRT group was approximately 12 years younger than the RT-only group. Given that younger age is a strong independent prognostic factor, this imbalance could have significantly influenced the observed survival trends. Disease-specific survival may provide a more precise understanding of the impact of chemoRT on MCC outcomes. If possible, the authors should include DSS data, with K-M curve, as it would help distinguish deaths due to MCC from other causes, particularly given the older age of the RT-only group.
Response:
We agree with the reviewer that the age difference could have influenced the observed survival trends. We attempted to control for these differences through multivariate analysis.
Unfortunately, we do not have DSS data. Nevertheless, we are adding the disease-free survival (DFS) data's K-M curve (Supplement 1). There was no significant difference in DFS between the groups.
Minor Comments
- Interpretation of Multivariable Analysis in Table 2
The multivariable analysis in Table 2 suggests that stage did not significantly impact OS (HR=0.7, p=0.43). This finding is unexpected, as prior studies have consistently shown that advanced stage is a major prognostic factor in MCC. If my understanding is correct, this discrepancy raises concerns about the validity of the model and should be discussed.
Response:
The population in our study contained only local/locally advanced resectable patients, eligible for radiotherapy, and no patients with metastatic disease. Thus, not all TNM components are included here. Furthermore, since these are relatively small cohorts, there is not enough statistical power to differentiate between these subgroups from the survival point of view.
We have also revised the multivariate analysis (Table 2) in the manuscript to better reflect our findings. The confidence interval for the TNM stage is quite broad (0.373-1.833), making it challenging to assess its impact on overall survival (OS). In larger cohorts with these specific characteristics, the stage may have significantly influenced OS outcomes.
- Details on Chemotherapy Regimens and ChemoRT Schedule
The manuscript does not specify which chemotherapy regimens were used or the timing of chemotherapy administration relative to radiation. Were chemotherapy and radiation given concurrently, sequentially, or variably across patients?
Response:
The chemotherapy regimens consisted of platinum, etoposide, and were given concurrently with radiation. We added this to the manuscript in line 173.
Sincerely,
Ronen Brenner, MD
ronenbr@wmc.gov.il
Ronen.brenner@gmail.com
Reviewer 2 Report
Comments and Suggestions for Authors
The authors report on a retrospective study on 105 patients diagnosed with MCC and treated with curative intend. The data is based on clinical practice from e-medical records in three medical centers from 1985 to 2024. Criteria for treatment by treating physicians’ discretion. Two cohorts were defined either by radiation alone or in combination with chemotherapy in the adjuvant setting. The main findings are no significant statistical differences in either PFS or OS rates at 20 years between the two cohorts, although there seem to be a numeric difference favoring adding chemotherapy that the authors recommend testing in a prospective trial.
The limitations of the study as mentioned by the authors are many due to the nature of a retrospective study, including the unbalanced patient groups and the need of matching. Since there seem to be a signal of effect of adding chemotherapy one should consider if the following factors could influence the outcome and taken into consideration in the paper:
Patients in the radiotherapy alone cohort were >11 years older (median age 65.9 [IQR 59-73.0] vs. 77.3 [IQR 61.4-83.3] years). Since the older patients have shorter life expectancy does this impact OS – and would a 20-year OS rate be considered too long (what is the mean life expectancy?) for this elderly group of patients? Would a 5 or even 10 year OS rate be more appropriate?
How were other known risk factors distributed: ecog performance status, co-morbidities, immune suppression?
Did the lower radiation doses to patients treated with radiotherapy alone (median 45 Gy vs 50 Gy) impact outcome?
What treatments were provided upon recurrence: equally between the two cohorts, type of systemic treatments (chemo- and/or immunotherapy)? This could impact OS significantly and should be taken into consideration in the paper.
What types of chemotherapy (fx platin based) and immunotherapies (anti PD-1/L1 based etc) were given?
As data from patients with metastatic MCC has not shown a survival benefit with chemotherapy but has increased with immunotherapy, how do the authors see the role of chemotherapy in the very promising era of immunotherapy, and how do they suggest this to be tested in a prospective study?
Author Response
Ronen Brenner, MD
Director of the Oncology Institute
Wolfson Medical Center
62 Halohamim Street, Holon 5822012, Israel
ronenbr@wmc.gov.il
(972)-54-3456721
28 February, 2025
Editor-in-Chief
Cancers
cancers@mdpi.com
Dear Editor and Reviewers,
Thank you for thoroughly reviewing our manuscript “Is the addition of chemotherapy to adjuvant radiation beneficial in Merkel cell cancer? A real-world data analysis with long-term follow-up”. We greatly appreciate your dedicated time and effort in providing comprehensive feedback. Your constructive comments and suggestions have greatly enhanced the content of our manuscript. We have carefully addressed all the points raised and revised our manuscript accordingly. Below, we provide a point-by-point response to each comment, with detailed explanations of the changes made. The modifications are also highlighted in the revised manuscript.
Response to Reviewer #3:
The limitations of the study as mentioned by the authors are many due to the nature of a retrospective study, including the unbalanced patient groups and the need of matching. Since there seem to be a signal of effect of adding chemotherapy one should consider if the following factors could influence the outcome and taken into consideration in the paper:
Comment 1
Patients in the radiotherapy alone cohort were >11 years older (median age 65.9 [IQR 59-73.0] vs. 77.3 [IQR 61.4-83.3] years). Since the older patients have shorter life expectancy does this impact OS – and would a 20-year OS rate be considered too long (what is the mean life expectancy?) for this elderly group of patients? Would a 5 or even 10 year OS rate be more appropriate?
Response:
We agree with this comment and have calculated overall survival and disease-free survival rates after 5 years of follow-up. The 5-year survival rates were quite similar between the groups: 64.9% in the chemoradiation group and 64.5% in the radiation group (p=0.89). Five-year disease-free survival rates were also similar, 57.4% in the chemoradiation group and 58.7% in the radiation group (p=0.80).( lines 180-184).
Comment 2
How were other known risk factors distributed: ecog performance status, co-morbidities, immune suppression?
Response:
We thank the reviewer for this comment, which we have added to the “limitations” section, lines 284-285. These are important risk factors; however, in this retrospective study, we did not have information on patients' performance status and comorbidities. It would be beneficial to collect this data in future prospective studies.
Comment 3
Did the lower radiation doses to patients treated with radiotherapy alone (median 45 Gy vs 50 Gy) impact outcome?
Response:
Thank you for this comment added also in the manuscript (lines 249-254). The lower radiation dose administered to patients receiving radiotherapy alone may have influenced the outcome. While the difference in radiation doses between the study groups was statistically significant, the actual numeric difference was slight—approximately 10%. This difference is likely less influential on the overall outcomes compared to the effects of chemotherapy.
Comment 4
What treatments were provided upon recurrence: equally between the two cohorts, type of systemic treatments (chemo- and/or immunotherapy)? This could impact OS significantly and should be taken into consideration in the paper.
What types of chemotherapy (fx platin based) and immunotherapies (anti PD-1/L1 based etc) were given?
Response:
We thank the reviewer for this comment, added also in the manuscript (lines 285-296). The information regarding the treatments provided upon recurrence was not available. However, we calculated the number of patients who theoretically could have received immunotherapy. Immunotherapy was introduced in the Israeli Health Benefits Package in 2018 and only for metastatic MCC. Since the patient population in this study had local MCC, the only patients who could have benefited from immunotherapy were those who had disease recurrence after 2018. There were seven such patients with recurrent disease during 2018-2023, with a similar distribution between the groups, three from the chemoradiation group and four from the radiation-only group. We do not have information on whether their disease recurrence was metastatic. Nevertheless, the small number of patients and the same distribution between the groups suggest that immunotherapy could not have influenced the outcome of this study.
All chemotherapy patients included in this study received platinum and etoposide.
Comment 5
As data from patients with metastatic MCC has not shown a survival benefit with chemotherapy but has increased with immunotherapy, how do the authors see the role of chemotherapy in the very promising era of immunotherapy, and how do they suggest this to be tested in a prospective study?
Response:
We thank the reviewer for this comment addressed also in the manuscript (lines 302-310). Unlike melanoma, MCC is responsive to chemotherapy. This indicates that the addition of chemotherapy to immunotherapy may have a synergistic effect on survival, similar to the case of SCLC, also a neuroendocrine carcinoma. This hypothesis should be tested in future prospective studies in appropriately characterized patients with metastatic disease and possibly also in neoadjuvant and adjuvant settings.
Sincerely,
Ronen Brenner, MD
ronenbr@wmc.gov.il
Ronen.brenner@gmail.com
Reviewer 3 Report
Comments and Suggestions for Authors
Authors have evaluated the long-term survival outcomes of 105 patients with MCC treated with surgery and radiation alone or combined chemoradiation between 1985-2023.
The 20-year overall survival rates were 53.4% for chemoradiation versus 30.7% for radiation alone (p=0.324). After controlling for age, stage, and tumor location in a multivariable analysis, survival differences were still not significantly different (Hazard ration (HR)=1.36, 95% CI 0.61-3.00, p=0.450).
I have the following comments:
1) Authors investigate 20-year survival; this means that MCC is a disease with a very good prognosis as in other diseases only part of patients survive 1, 3 or 5 years. I think, the main problem here is: after 10 or 15 or 20 years they do not die from cancer anymore but from other diseases. So this big difference for overall survival rates were 53.4% for chemoradiation versus 30.7% for radiation alone is not clear for me. I think that 5- or 10-year survival would rather show the impact of cancer therapy than 20-year survival (and after 5 years, there is no difference between groups as KM curves show).
2) Patient samples are very small resulting in lack of significant values. I do not recommend overvaluing the significance when analyzing only 50 patiets. Better discuss clinical relevance of differences, whereby there are no differences after 5 years.
3) I can see ‘without glasses’ that KM curves cross and Hazard assumption is not satisfied here. You cannot use Cox proportional hazard regression but need to switch to alternative methods like time depending Cox or even logistic regression with proportion of survival after 5, 10 and so on years (yes vs. no as depending on variable).
4) You analyzed long time period starting in 1985. In this time, therapy was developed, and new therapies were available; even chemotherapy changed including further drugs with better effectiveness and less side effects. How the time period of the study impacted results. If you analyze 5-year survival in patients included in 1985-1995 versus 2010-2015, for example?
Author Response
Ronen Brenner, MD
Director of the Oncology Institute
Wolfson Medical Center
62 Halohamim Street, Holon 5822012, Israel
ronenbr@wmc.gov.il
(972)-54-3456721
28 February, 2025
Editor-in-Chief
Cancers
cancers@mdpi.com
Dear Editor and Reviewers,
Thank you for thoroughly reviewing our manuscript “Is the addition of chemotherapy to adjuvant radiation beneficial in Merkel cell cancer? A real-world data analysis with long-term follow-up”. We greatly appreciate your dedicated time and effort in providing comprehensive feedback. Your constructive comments and suggestions have greatly enhanced the content of our manuscript. We have carefully addressed all the points raised and revised our manuscript accordingly. Below, we provide a point-by-point response to each comment, with detailed explanations of the changes made. The modifications are also highlighted in the revised manuscript.
Response to Reviewer #2:
Comment 1
Authors investigate 20-year survival; this means that MCC is a disease with a very good prognosis as in other diseases, only part of patients survive 1, 3, or 5 years. I think the main problem here is that after 10, 15, or 20 years, they do not die from cancer anymore but from other diseases. So, this big difference in overall survival rates, 53.4% for chemoradiation versus 30.7% for radiation alone, is not clear to me. I think that 5- or 10-year survival would rather show the impact of cancer therapy than 20-year survival (and after 5 years, there is no difference between groups, as KM curves show).
Response:
We thank the reviewer for this valuable comment. The reviewer mentions that the disease-free KM curve (added to the revised manuscript) opens after 5 years. This could be due to death from other diseases or the advanced age of the patients.
We followed the reviewer’s suggestion and calculated overall survival and disease-free survival rates after 5 years of follow-up. The 5-year survival rates were quite similar between the groups: 64.9% in the chemoradiation group and 64.5% in the radiation group (p=0.89). Five-year disease-free survival rates were 57.4% in the chemoradiation group and 58.7% in the radiation group (p=0.80).
Comment 2
Patient samples are very small resulting in lack of significant values. I do not recommend overvaluing the significance when analyzing only 50 patiets. Better discuss clinical relevance of differences, whereby there are no differences after 5 years.
Response:
We agree with the reviewer. The sample size was calculated using significant levels of 5% and power of 80%, assuming the same sample size in both groups. We presumed that the average probability of survival to the end of follow-up (20 years) is 30%. Using a hazard ratio (HR) of 0.5, 102 patients were needed, i.e., 51 in each group. We added this comment to the article (lines 154-158). Despite the small sample size, there is clinical relevance indicating that in advanced patients, whenever possible, it is useful to add chemotherapy to radiation.
Comment 3
I can see ‘without glasses’ that KM curves cross and Hazard assumption is not satisfied here. You cannot use Cox proportional hazard regression but need to switch to alternative methods like time-dependent Cox or even logistic regression with the proportion of survival after 5, 10, and so on years (yes vs. no as depending on the variable).
Response:
We thank the reviewer for this valuable comment that enhances the quality of the article. During the first period of the 20 years OS, approximately the first 5 years, the KM curves are almost adjacent, and then they open. This and the lack of significance in univariate and multivariate analyses indicate that the violation of the proportional hazard assumption is very small. Furthermore, we have followed the reviewer’s suggestion and analyzed overall survival during the first five years of follow-up. In addition to the univariate analysis described in the manuscript, lines 179-186, we also calculated the 5-year multivariable Cox regression. The results are presented below and in Table 2 in the revised article. Except for age at diagnosis, none of the variables influenced mortality.
Table 1: Multivariate analysis of 5 years overall survival
|
Factor |
HR |
95%CI |
p value |
|
Chemotherapy |
1.547 |
0.656-3.65 |
0.319 |
|
Age at diagnosis |
1.064 |
1.029-1.100 |
<0.001 |
|
TNM stage |
0.696 |
||
|
I |
ref. |
||
|
II |
1.114 |
0.385-3.229 |
|
|
III |
1.492 |
0.570-3.904 |
|
|
Location |
0.113 |
||
|
Head&neck |
ref. |
||
|
Rest of the body |
0.452 |
0.213-0.960 |
|
|
Unknown primary |
0.517 |
0.173-1.546 |
Comment 4
You analyzed long time period starting in 1985. In this time, therapy was developed, and new therapies were available; even chemotherapy changed including further drugs with better effectiveness and less side effects. How the time period of the study impacted results. If you analyze 5-year survival in patients included in 1985-1995 versus 2010-2015, for example?
Response:
We appreciate the reviewer's insightful feedback, which was also included in the updated version (lines 289-296). Immunotherapy was indeed the most significant change in the treatment of MCC over the research period. Since 2018, the Israeli Health Benefits Package included immunotherapy for MCC, but only for metastatic MCC. Since the study’s patient cohort had local MCC, only individuals who experienced a disease recurrence after 2018 could have benefited from immunotherapy. There were seven such patients with recurrent disease during 2018-2023, with a similar distribution between the groups, three from the chemoradiation group and four from the radiation-only group. We do not have information on whether their disease recurrence was metastatic. Nevertheless, the small number of patients with recurrence and the same distribution between the groups suggest that immunotherapy could not have influenced the outcome of this study.
We have also analyzed 5-year survival and reported that the survival rates were quite similar between the groups: 64.9% in the chemoradiation group and 64.5% in the radiation group (p=0.89) (lines 180-184 in the article).
Sincerely,
Ronen Brenner, MD
ronenbr@wmc.gov.il
Ronen.brenner@gmail.com
Round 2
Reviewer 1 Report
Comments and Suggestions for Authors
The authors appropriately addressed the comments and suggestions.
Reviewer 2 Report
Comments and Suggestions for Authors
No further comments
Reviewer 3 Report
Comments and Suggestions for Authors
-